# The effect of dumping syndrome severity on health-related quality of life among bariatric surgery patients in Saudi Arabia

**Leena R. Baghdadi** [1,2*], **Hoor K. Aloraini**[3], **Razan A. Almohanna**[3], **Jana I. Alhazmi**[3], **Farah M. Alhalafi**[3], **Sarah H. Alotaebe**[3]

**1** Department of Family and Community Medicine, College of Medicine, King Saud University, Riyadh, Saudi Arabia, **2** Department of Family and Community Medicine, King Saud University Medical City, Riyadh, Saudi Arabia, **3** College of Medicine, King Saud University, Riyadh, Saudi Arabia

* lbaghdadi@ksu.edu.sa

## Abstract

Dumping syndrome is a postoperative complication of bariatric surgery; it is classified into early and late dumping syndrome depending on the time of the symptoms. This study aimed to assess the health-related quality of life among adult patients in Saudi Arabia experiencing varied severity of dumping syndrome after a sleeve gastrectomy or a Roux-en-Y gastric bypass procedure. This cross-sectional study was conducted using a validated health-related quality of life questionnaire administered online to adults (≥17 years old), who had undergone bariatric surgery in Saudi Arabia. The study recruited 237 eligible adults; 142 (59.9%) of them had severe dumping syndrome. The mean physical health component summary score was 45.01 (SD = 8.42) and the mean mental health component summary score was 38.89 (SD = 9.74). Individuals with severe dumping syndrome had significantly lower (p = 0.041) physical and mental health component summary scores compared to individuals with moderate severity dumping syndrome. Higher household income was associated with higher physical and mental health summary scores. Participants who underwent gastric bypass surgery had significantly lower physical health summary scores compared to those who underwent sleeve gastrectomy. Dumping syndrome is more prevalent in women than men. The most common symptoms are a desire to lie down (79.3%) and nausea (37.9%). Patients with severe late dumping syndrome were more inclined to seek help from various healthcare providers, highlighting potential gaps in effective management strategies. These findings suggest current management approaches for severe late dumping syndrome may be insufficient, prompting the need for developing standardized treatment protocols and increasing awareness among healthcare providers. Future research should focus on evaluating the effectiveness of different therapeutic interventions and exploring strategies to improve patient outcomes and access to specialized care.

**Data availability statement:** All data are in the manuscript and/or supporting information files.

**Funding:** The author(s) received no specific funding for this work.

**Competing interests:** The authors have declared that no competing interests exist.

## Introduction

Since 1980, the prevalence of obesity has gradually increased worldwide, highlighting the growing need for effective treatment strategies [1]. Bariatric surgery has emerged as a highly effective intervention for morbid obesity and its associated comorbidities, outperforming non-surgical approaches [2,3]. Among the currently available surgical procedures, sleeve gastrectomy (SG) and Roux-en-Y gastric bypass (RYGB) are most common worldwide [4].

Despite their efficacy, bariatric surgeries carry a risk of postoperative complications. Dumping syndrome (DS) is a commonly reported complication following gastric or bariatric procedures. [5]. It is further classified into early and late DS depending on the time of the symptoms (within 1 hour and 1–3 hours respectively). In Saudi Arabia, Baghdadi et al (2025) reported a high prevalence of DS (67.3%), with early DS accounting for 76.8% of cases and late DS representing 23.2% [6] of participants. Patients experiencing severe early and moderate late DS symptoms showed greater deterioration across multiple aspects of health-related quality of life (HRQoL) [7].

Understanding DS is crucial, because its effects not only influence the surgical outcomes, but also have significant psychological and practical effects on the person's quality of life, as it markedly reduces HRQoL, in particular [8]. A recent longitudinal analysis has shown a significant association between DS (especially the late phase) and eating disorders, specifically binge eating and social withdrawal with higher levels of anxiety [9]. A retrospective cohort study also found that early DS after SG is common and associated with poorer HRQoL, particularly in the domains of mental and general health, emotional well-being, and vitality [10].

Despite the results of these studies, few studies have directly compared the severity, quality of life, and general health condition between the early and late phases of DS. A systematic review highlighted a lack of consensus on diagnostic tools and minimal data on how each phase of DS uniquely affects patients' lives [11]. We found a single study that focused on QoL among 400 post-bariatric surgery patients across various regions in Saudi Arabia, where 35% of participants reported a fair quality of life, and 25% experienced poor quality of life [12]. Therefore, we aimed to further investigate the quality of life for patients with DS by including a local Saudi study population and more than one type of gastric surgery, to achieve a comprehensive comparison in the intensity of early and late stages of DS and their effects on HRQoL among Saudi individuals who have undergone SG or gastric bypass surgery.

## Materials and methods

### Study sample and study design

This cross-sectional study was conducted using a validated HRQoL questionnaire administered online to adults who had undergone bariatric surgery in Saudi Arabia. Data was collected from February 2024 until December 2024. The inclusion criteria were individuals aged ≥17 years, who live in Saudi Arabia and had undergone bariatric surgery—specifically SG or RYGB, collectively referred to by the IFSO as 'Metabolic and Bariatric Surgery (MBS)" at least 3 months ago to capture early

manifestations of DS and allow for timely interventions. The exclusion criteria were individuals undergoing bariatric procedures other than SG or RYGB and residing outside Saudi Arabia.

Based on a study conducted at King Fahad General Hospital in Jeddah, which reported a 31.4% prevalence of DS [13], we used a single proportion formula with 95% level of confidence and 5% margin of error, and estimated the sample size to be 331.

$$sample\ size: \frac{(Z_{\alpha/2})^2 \times p(1-p)}{d^2} = \frac{1.96^2 \times 0.314(1-0.314)}{0.05^2} = 331$$

## Questionnaires

**Modified sigstad dumping syndrome questionnaire.** This was adapted from the original Sigstad score system [14], which had 16 items with a threshold of ≥7 points. The modified version had 10 items with a diagnostic threshold of ≥3.26 points. Individuals who exceeded this threshold were considered to have DS and were further classified into early and late DS groups based on the onset of symptoms. If symptoms occurred within 1 hour postprandially, it was early DS, and if symptoms occurred 1–3 hours postprandially, it meant late DS [15]. To measure the intensity of each symptom, a questionnaire with a 4-point Likert scale of intensity (not at all, a little, quite a bit, and very much) was used. The severity rating of DS (low, moderate, and severe) was computed based on the OSCAR dumping syndrome severity questionnaire, these criteria were provided by Anandavadivelan et al (2020) [7].

To measure symptom intensity, a question about the troublesomeness of each symptom was asked, with a 4-point Likert scale: not at all, a little, quite a bit, and very much. Trouble or troublesome meant the intensity of symptoms that were annoying or affected daily activity. For the severity rating of DS, low meant no symptom that is troublesome, moderate meant (a little bit to quite a bit) at least one symptom that is little to a bit troublesome, and severe (very much): at least two symptoms with quite a bit to very much troublesome.

**SF-12 questionnaire.** The SF-12 [16] was adapted from the SF-36 HRQoL [17]. The number of items were decreased from 36 to 12, to create a more convenient version. It is a self-assessment survey that assesses eight domains of functional health and well-being: physical functioning, role limitations due to physical problems, pain interference, general health, vitality, social functioning, role limitations due to emotional problems, and mental health. Scores are calculated for each of these eight domains of the SF-12, in addition to two summary scores, the physical health component summary (PHS) and mental health component summary (MHS). In this study, we used the translated Arabic version of the questionnaire [18]. The HRQol physical health and mental health components were computed using regression weights provided by the scoring method for the SF-12 Questionnaire [16] by transforming the questionnaire into indicator variables and using the regression weights to adjust these indicator variables, then summing them into two main linearized norm-based components (physical and mental health summary components). The eight domains of the HRQoL were computed by transforming them into 0–100 scales. All the HRQol questionnaire items were reverse scored so a higher score denoted better health before computing the physical and mental health components.

## Ethics

All data collected were kept confidential. Prior to the study, participants gave written informed consent in Arabic and the study's purpose was thoroughly explained to the participants. They were informed that their responses were voluntary and that they could withdraw at any time without any obligation to the research team. The information provided was anonymous, with participants assigned unique identification numbers to ensure privacy. No incentives or rewards were offered, and participation was entirely voluntary. The study received ethics approval from the Institutional Review Board at the College of Medicine, King Saud University (approval number: E-24–8508; date of approval: January 21, 2024).

## Statistical analysis

Descriptive analysis with the mean and standard deviation were applied to continuous variables, and frequencies and percentages for categorical data. The Kolmogorov-Smirnov test was used to assess the normality assumption for metric variables along with histograms. The independent samples t-test was used to assess the statistical significance of mean differences in numerical variables between two subgroups (early DS vs late DS) while a one-way analysis of variance (ANOVA) test was used to compare means across three categories (levels of DS symptoms severity). The Chi-square test of independence or Fisher's exact test (when counts were small) were used to assess the association between two categorical variables (or the difference between subgroups with regards to categorical outcomes). Multivariable binary logistic regression analysis was applied to assess the statistical significance of predictors for the bariatric surgery patients' odds of having DS, the association between independent predictor variables with the analyzed outcome was expressed as multivariable adjusted odds ratios and corresponding 95% confidence intervals (CI). Multivariable linear regression analysis was applied to numerical outcomes using patients' sociodemographic as well as health-related predictors. The results of linear regression were reported as unstandardized beta coefficients with their associated 95% CI. SPSS Statistics software (v.30 IBM Corp., Armonk, NY, USA) was used for all descriptive and inferential analysis. P values <0.05 were reported as statistically significant.

## Results

Data were collected between February 2024 and December 2024 via the validated online questionnaires. A total of 373 bariatric surgery patients agreed to participate and completed the questionnaire. However, 21 participants were excluded from the study because they did not meet the inclusion criteria related to the type of bariatric surgery performed or the duration since the surgery (specifically, surgeries must have been conducted at least three months prior to ensure the capture of early manifestations of dumping syndrome and facilitate timely interventions). Additionally, participants had to be at least 17 years old to qualify. A total of 352 eligible participants completed the questionnaire, which met the required sample size for the study. However, only 237 of these participants had a modified Sigstad weighted DS score of ≥3.26 indicating the presence of DS; and the remaining 115 participants with modified Sigstad weighted DS scores of <3.26 have been excluded. The eligible 237 participants were further divided into two groups; 182 (76.8%) of them had early DS (symptoms within 1 hour postprandially) and 55 (23.2%) had late DS (symptoms 1–3 hours postprandially).

The study participants were asked to rate the intensity of their DS symptoms using a 4-point Likert scale; with a higher score implying a higher intensity of the symptoms (Fig 1). Based on these scores, we found that 3.4% of the participants had experienced very low intensity DS symptoms, 36.7% of them experienced moderate intensity symptoms, and 59.9% had severe intensity DS symptoms. However, no significant differences were found between the early DS and late DS subgroups, with regards to overall perceived troubles and severity of symptoms.

Table 1 shows that of the 237 participants, 75.1% were women and 24.9% were men. The monthly household income (HHI) of 55.3% of respondents was ≤ 5000 SAR/month. Most participants (93.2%) underwent SG, while 6.8% had RYGB surgery. The majority of participants (80.2%) complied with attending the postoperative follow-up sessions. Among the non-compliant patients (19.8%), the main reason for not attending the postoperative sessions was unwillingness (36.2%). The analysis showed that women on average have significantly higher PHS and MHS scores compared to men (4.12 units higher, p = 0.011, and 4.53 units higher, p = 0.012, respectively). Participants with higher HHI have significantly higher PHS (p = 0.004) and score 1.69 units per additional 5000 SAR/month. Households with higher incomes also reported higher MHS (p < 0.001). Each additional 5000 SAR/month was associated with 2.25 units increase in the MHS. According to the multivariable analysis, the bariatric surgery patients who underwent RYGB surgery had significantly lower PHS scores compared to those who underwent SG surgery (4.99 units lower on average, p = 0.019). The patients who were non-compliant with post-bariatric surgery appointments due to time constraints had a significantly lower PHS score compared to those patients who had no time constraints (4.69 units lower on average, p = 0.015).

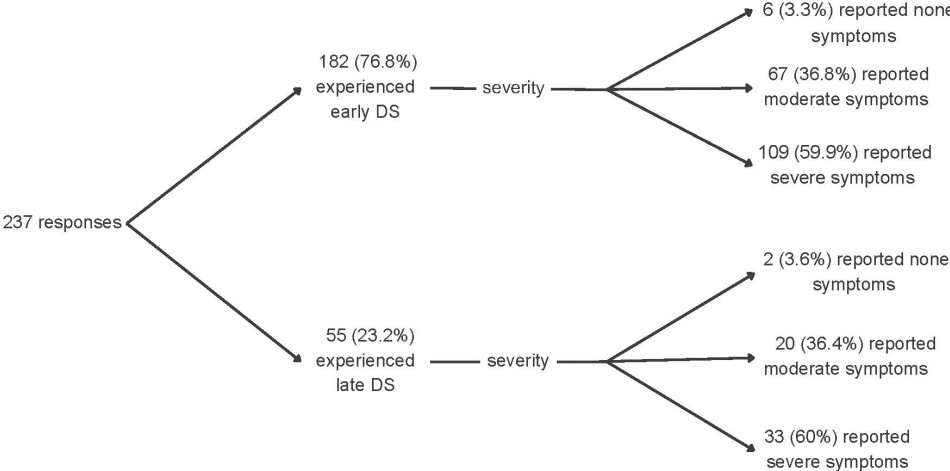

**Fig 1. Distribution of early and late Dumping Syndrome Symptoms among 237 participants.** None-very low: No symptoms that were troublesome; Moderate: at least one symptom that was little to a bit troublesome; Severe: at least two symptoms with quite a bit to very much troublesome.

Table 2 shows the majority of patients (67.5%) had intervened their symptoms, with an approaching significance (p = 0.054) of a higher proportion of patients with late DS (78.2%) compared to patients with early DS (64.3%). Among those who intervened, 62.5% visited the surgeons (who had performed their operations) for post-surgical visits, 60.0% of participants followed non-medical management methods to alleviate their DS symptoms. A significantly greater percentage of patients in the late DS group (58.1%) sought medical help by visiting another physician, compared to 29.1% in the early DS group; this difference was statistically significant (p < 0.001). Some participants (52.3%) had received health education regarding DS before surgery. Having previous health education regarding DS was associated with higher a MHS (2.27 units higher); however, the relationship is marginally significant (p = 0.061).

Table 3 shows the HRQoL scores and its various aspects. According to the linear weighted HRQoL questionnaire self-rating, the participants' PHS score mean was 45.01 (SD = 8.42), and the MHS score mean was 38.89 (SD = 9.74). The highest HRQoL subdomain score was reported for the general health domain (mean 58.48 [SD = 21.34]) while the lowest score was for vitality (mean 41.60 [SD = 20.81]). Late DS patients have significantly higher mental health scores (mean score = 53.64) compared to early DS patients (M = 47.91), p = 0.041. Significant differences between DS severity levels were found for HRQoL, PHS and MHS scores; patients with severe DS had significantly lower scores compared to patients in moderate severity subgroups. Perceived troubles and interference (TIN) with the DS score had a significant negative association with MHS scores (p = 0.018). Higher TIN scores were associated with lower MHS scores, holding other factors constant. On average, an increase of one unit in the TIN score was associated with 1.72 units reduction in the MHS score (S1 Table).

S1 Table. Descriptive analysis of the patients perceived interference and troubles associated with DS symptoms (n = 237).

Fig 2 shows that the PHS score increases with increasing severity of DS and the highest intensity of symptoms have a higher mean PHS score than with low intensity symptoms. Whereas there is a steady decline in the MHS scores with increasing intensity of DS symptoms. MHS was lowest for the severe DS symptoms group with a mean of 36.71 (SD = 9.58), the lowest PHS was also in the severe DS symptoms group with a mean of 43.90 (SD = 8.86). The analysis model findings showed that the perceived PHS score had correlated significantly and negatively with their odds of having DS. For each additional one-point rise in the patients' mean perceived PHS score, their odds of having DS decreased by a factor of 5.1% times less on average, p = 0.006. Similarly, the patients' mean perceived MHS score correlated negatively

**Table 1. Association between bariatric patients' sociodemographic characteristics and early and late dumping syndrome symptom severity (n = 237).**

| | Early dumping syndrome: Severity of symptoms | | | Late dumping syndrome: Severity of symptoms | | | Comparison between six subcategories |
|---|---|---|---|---|---|---|---|
| | **None-Very Low**[1] | **Moderate**[2] | **Severe**[3] | **None-Very Low**[1] | **Moderate**[2] | **Severe**[3] | |
| **Sex** | | | | | | | p = 0.060[F] |
| Female | 5 (83.3%) | 1 (76.1%) | 88 (80.7%) | 1 (50.0%) | 15 (75.0%) | 18 (54.5%) | |
| Male | 1 (16.7%) | 16 (23.9%) | 21 (19.3%) | 1 (50.0%) | 5 (25.0%) | 15 (45.5%) | |
| **Age (years)** | 38.17 ± 8.47 | 31.93 ± 9.11 | 31.04 ± 10.29 | 30.00 ± 5.66 | 32.95 ± 8.04 | 30.67 ± 7.55 | p = 0.522[A] |
| **Household monthly income (Saudi Riyals)** | | | | | | | p = 0.275[F] |
| <5000 SAR/month | 2 (33.3%) | 37 (55.2%) | 69 (63.3%) | 1 (50.0%) | 8 (40.0%) | 14 (42.4%) | |
| 5000-10000 SAR/month | 2 (33.3%) | 14 (20.9%) | 21 (19.3%) | 1 (50.0%) | 7 (35.0%) | 11 (33.3%) | |
| 11000-15000 SAR/month | 1 (16.7%) | 7 (10.4%) | 4 (3.7%) | 0 (0%) | 4 (20.0%) | 2 (6.1%) | |
| >15000 SAR/month | 1 (16.7%) | 9 (13.4%) | 15 (13.8%) | 0 (0%) | 1 (5.0%) | 6 (18.2%) | |
| **Residence in province** | | | | | | | p = 0.357[F] |
| Eastern Provinces | 0 (0%) | 12 (17.9%) | 25 (22.9%) | 1 (50.0%) | 6 (30.0%) | 6 (18.2%) | |
| Western Provinces | 4 (66.7%) | 9 (13.4%) | 20 (18.3%) | 0 (0%) | 5 (25.0%) | 5 (15.2%) | |
| Northern Provinces | 0 (0%) | 8 (11.9%) | 14 (12.8%) | 0 (0%) | 0 (0%) | 2 (6.1%) | |
| Southern Provinces | 0 (0%) | 7 (10.4%) | 11 (10.1%) | 0 (0%) | 2 (10.0%) | 2 (6.1%) | |
| Central Region | 2 (33.3%) | 31 (46.3%) | 39 (35.8%) | 1 (50.0%) | 7 (35.0%) | 18 (54.5%) | |
| **Type of bariatric surgery** | | | | | | | p = 0.662[F] |
| Sleeve Gastrectomy | 6 (100%) | 63 (94.0%) | 103 (94.5%) | 2 (100%) | 18 (90.0%) | 29 (87.9%) | |
| Roux-en-Y gastric bypass | 0 (0%) | 4 (6.0%) | 6 (5.5%) | 0 (0%) | 2 (10.0%) | 4 (12.1%) | |
| **Attended post-operation follow-up sessions** | | | | | | | p = 0.151[F] |
| No | 1 (16.7%) | 15 (22.4%) | 16 (14.7%) | 1 (50.0%) | 3 (15.0%) | 11 (33.3%) | |
| Yes | 5 (83.3%) | 52 (77.6%) | 93 (85.3%) | 1 (50.0%) | 17 (85.0%) | 22 (66.7%) | |
| **Reasons for non-compliance at appointments** | (n = 1) | (n = 15) | (n = 16) | (n = 1) | (n = 3) | (n = 11) | p = 0.967[F] |
| Financial | 0 (0%) | 6 (40.0%) | 5 (31.3%) | 0 (0%) | 1 (33.3%) | 3 (27.3%) | p = 0.622[F] |
| Transportation | 0 (0%) | 4 (26.7%) | 6 (37.5%) | 1 (100%) | 0 (0%) | 4 (36.4%) | p = 0.823[F] |
| Time constraint | 0 (0%) | 4 (26.7%) | 5 (31.3%) | 1 (100%) | 1 (33.3%) | 4 (36.4%) | p = 0.383[F] |
| Unwillingness | 1 (100%) | 6 (40.0%) | 5 (31.3%) | 0 (0%) | 1 (33.3%) | 4 (36.4%) | |

Reported values are frequency (%) or Mean ± standard deviation; [F] Fisher's Exact test, [A] one-way analysis of variance (ANOVA), [1] None-very low: No symptoms that is troublesome; [2] Moderate: at least one symptom that is little to a bit troublesome; [3] Severe: at least two symptoms with quite a bit to very much troublesome

and statistically significantly (p < 0.001) with their odds of having DS. For each additional one-point rise in the patients mean perceived MHS score, their mean predicted odds of having DS declined by a factor of 5.8% times less on average, p < 0.001. These findings indicate that patients with DS may experience lower quality of physical and mental health in general.

## Discussion

This study investigated the quality of life for patients with DS by including a local Saudi study population and more than one type of gastric surgery, to achieve a comprehensive comparison in the intensity of early and late stages of DS and their effects on HRQoL among Saudi individuals who have undergone SG or gastric bypass surgery. Study participants exhibited a mean PHS score of 45.01 (SD = 8.42) and a mean MHS score of 38.89 (SD = 9.74), highlighting a substantial burden on both physical and mental well-being. Those with severe DS experienced significantly lower scores in both domains compared to individuals with moderate severity, highlighting the profound impact of symptom intensity on the

**Table 2. Early and late dumping syndrome severity and intervention (n = 237).**

| | Patient inter-vened symptoms | Intervention undertaken | | | | Patient had health education on DS |
| --- | --- | --- | --- | --- | --- | --- |
| | | Visited surgeon who performed operation | Visited another physician | Went to the pharmacist | Non-medical methods (inter-net, family and friends etc.) | |
| **Early DS** | | | | | | |
| Severity of symptoms | | | | | | |
| None-Very Low[1] | 5 (83.3%) | 4 (80.0%) | 2 (40.0%) | 0 (0%) | 1 (20.0%) | 5 (83.3%) |
| Moderate[2] | 40 (59.7%) | 22 (55.0%) | 11 (27.5%) | 11 (27.5%) | 22 (55.0%) | 37 (55.2%) |
| Severe[3] | 72 (66.1%) | 44 (61.1%) | 21 (29.2%) | 18 (25.0%) | 46 (63.9%) | 54 (49.5%) |
| **Late DS** | | | | | | |
| Severity of symptoms | | | | | | |
| None-Very Low[1] | 2 (100%) | 1 (50.0%) | 1 (50.0%) | 0 (0%) | 1 (50.0%) | 2 (100%) |
| Moderate[2] | 15 (75.0%) | 9 (60.0%) | 9 (60.0%) | 5 (33.3%) | 10 (66.7%) | 10 (50.0%) |
| Severe[3] | 26 (78.8%) | 20 (76.9%) | 15 (57.7%) | 7 (26.9%) | 16 (61.5%) | 16 (48.5%) |
| **Comparison between six subcategories** | p = 0.362[F] | p = 0.509[F] | p = 0.024[F] | p = 0.826[F] | p = 0.464[F] | p = 0.494[F] |

DS, dumping syndrome; reported values are frequency (%), [F] Fisher's Exact test, [1] None-very low: No symptom that is troublesome; [2]Moderate: at least one symptom that is little to a bit troublesome; [3] Severe: at least two symptoms with quite a bit to very much troublesome.

quality of life. In our previous study [6], we found a high prevalence of DS (67.3%) among post-bariatric surgery patients in Saudi Arabia. Also, early DS was more common than late DS in this patient population. The findings from the current research and our previous study [6] have notable similarities. Our analysis revealed that there were no statistically significant associations between patients with early and late DS and the various sociodemographic factors, including sex, age, household income, and private health insurance. The participants in this study were predominantly women (75.1%), with a near equal distribution across the different DS subtypes, and there were no significant differences based on sex. This outcome is consistent with the findings of Alsulami et al and Banerjee et al, who also identified a higher prevalence of DS among females [19,20].

In this study, 93.2% of respondents underwent SG, while only 6.8% underwent RYGB surgery. Among them, 182 individuals (76.8%) experienced early DS, with symptoms occurring within one hour after eating, while 55 individuals (23.2%) experienced late DS, with symptoms appearing 1–3 hours after eating. Tack et al [21] reported a higher incidence of DS symptoms primarily among patients who underwent RYGB, with approximately 40% of patients experiencing related symptoms. This contrast in findings highlights the differing outcomes associated with various surgical procedures. While our study indicates a predominance of SG procedures, it prompts further exploration into the potential long-term outcomes and symptom profiles of RYGB patients, emphasizing the importance of understanding the implications of each surgical option [21].

The study revealed that a significant proportion of respondents (52.3%) had received prior health education regarding DS, with similar trends observed in both early and late DS groups. However, these findings contrast with other research. Rodgers et al [22] indicate that many people who undergo upper gastrointestinal surgery are not aware of DS and are not adequately educated about it [22].

This study highlights insights regarding the intensity of symptoms in early versus late DS. From the group of early DS participants, only 3.3% reported none or very low symptoms, while 59.9% experienced severe symptoms. In the late DS group, 3.6% of participants reported none or very low symptoms and 60% experienced severe symptoms. The factors contributing to symptom intensity include a prevalent desire to lie down (79.3%) and frequent reports of nausea (37.9%). Patients with early DS are more likely to experience nausea and abdominal cramps, while those with late DS have more severe symptoms, including fatigue and confusion. These differences suggest the importance of monitoring symptoms closely in patients with different types of DS.

**Table 3. Descriptive analysis of the patients' overall perceptions of health-related quality of life (n = 237).**

| | Total sample (n = 237) Mean±SD [range] | Early DS (n = 182) Mean±SD [range] | Late DS (n = 55) Mean±SD [range] | Comparison between subgroups |
|---|---|---|---|---|
| Linear weighted physical health component summary (PHS) score [0–100 range]* | 45.01 ± 8.42 [20.74 – 60.10] | 45.07 ± 8.61 [20.74 – 60.10] | 44.80 ± 7.84 [27.59 – 57.60] | p = 0.837[T] |
| Linear weighted mental health component summary (MHS) score [0–100 range]* | 38.89 ± 9.74 [16.09 – 57.81] | 38.47 ± 9.92 [16.09 – 57.81] | 40.28 ± 9.06 [21.35 – 57.31] | p = 0.229[T] |
| Overall total weighted health-related quality of life score [0–200 range] | 83.50 ± 11.50 [51.27 – 100.57] | 83.19 ± 11.19 [51.27 – 100.57] | 84.53 ± 12.55 [55.03 – 100.16] | p = 0.449[T] |
| Health-related quality of life sub-domains (expressed as normed %)** [0–100 range] | | | | |
| General health domain score (%) | 58.48 ± 21.34 [0–80] | 57.91 ± 21.49 [0–80] | 60.36 ± 20.90 [0–80] | p = 0.456[T] |
| Physical functioning domain score (%) | 44.66 ± 24.40 [0 – 66.67] | 44.78 ± 24.50 [0 – 66.67] | 44.24 ± 24.26 [0 – 66.67] | p = 0.886[T] |
| Role physical domain score (%) | 57.38 ± 44.61 [0–100] | 58.79 ± 43.72 [0–100] | 52.73 ± 47.55 [0–100] | p = 0.378[T] |
| Pain interference domain score | 51.48 ± 22.14 [0–75] | 50.82 ± 21.65 [0–75] | 53.64 ± 23.77 [0–75] | p = 0.410[T] |
| Vitality domain score | 41.60 ± 20.81 [0–80] | 41.43 ± 20.92 [0–80] | 42.18 ± 20.61 [0–80] | p = 0.815[T] |
| Social functioning domain score | 45.57 ± 27.73 [0–80] | 44.29 ± 28.00 [0–80] | 49.82 ± 26.63 [0–80] | p = 0.195[T] |
| Role emotional domain score | 56.33 ± 44.65 [0–100] | 56.04 ± 44.50 [0–100] | 57.27 ± 45.56 [0–100] | p = 0.858[T] |
| Mental health domain score | 49.24 ± 18.21 [0–80] | 47.91 ± 18.57 [0–80] | 53.64 ± 16.37 [20–80] | p = 0.041[T] |

DS· dumping syndrome, [T] independent samples t-test; *components are linear weighted factor-based scores using regression coefficients provided by John E Ware et al; ** Normed scores are non-weighted original scores computed by the summation method and rescaled as scores between 0–100 points subtracting the minimum possible value for these domains from the raw scores and dividing the result by the maximum possible score for each of those items then multiplying the product by 100.

We examined the behavior of participants with different intensities of early and late DS, in seeking help about their symptoms. The study's participants had intervened when they encountered the DS symptoms (62.5%). More participants with late DS (78.2%) sought help about their symptoms than the participants with early DS (64.3%). From the participants who took steps to manage their symptoms by either revisiting their operating surgeon, visiting another treating physician, a pharmacist or reaching out to non-medical methods (including the internet, family and friends etc.), more than half of the participants preferred to revisit the operating surgeon. However, patients with late DS, significantly (p < 0.001) preferred to visit another physician compared to patients with early DS. This could be attributed to the extent of the symptoms of late DS compared to early DS, as late DS shows symptoms related to neuroglycopenia, such as fatigue, weakness, confusion and syncope, that last 1–3 hours, and may affect cognitive function [5], and occur later than the symptoms experienced in early DS (within an hour postprandially) [21,23]. Thus, patients may feel an increased need to seek out treatment from another non-surgical physician.

When we considered health education about DS for participants from their physicians, we found that nearly half (52.3%) of the participants had received health education about DS from their physicians, with similar proportions among early (52.7%) and late (50.9%) DS participants. Although not statistically significant (p = 0.061), our data indicates that health education regarding DS before gastric surgery is associated with a higher MHS, reflecting an increase of

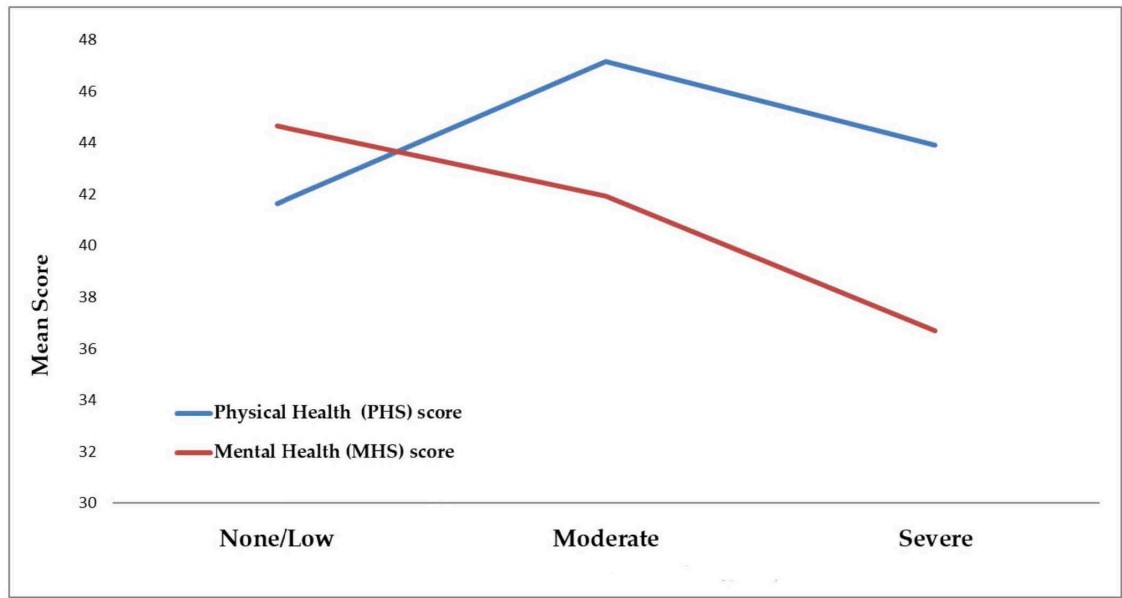

**Fig 2. Association between the severity of DS in bariatric surgery patients and their mental and physical health scores.**

approximately 2.27 units. These findings suggest that access to health education may positively influence mental health outcomes, emphasizing the potential benefits of educational initiatives in the realm of clinical settings.

Our study shows clear differences in HRQoL among bariatric surgery patients who experience varying severity levels of DS, where patients with severe DS reported markedly lower PHS and MHS compared to those with moderate symptoms. Multivariable linear regression analyses revealed that the presence of DS is a strong negative predictor of PHS and MHS scores (p<0.001). This aligns with prior literature demonstrating reduced HRQoL [5,8,24–29] and specifically diminished mental HRQoL post-surgery for patients experiencing DS, consistent with findings that DS contributes substantially to post-surgical anxiety and depression [9]. The findings from a retrospective cohort suggested insignificant relationships between the presence and severity of DS after RYGB and adequate postoperative weight loss [30]. However, our pattern of worse HRQoL for patients reporting severe early DS mirrors prospective cohort evidence from esophageal cancer surgery populations, where self-reported early and late DS predicted poorer HRQoL, particularly for those with severe early DS or moderate late DS symptoms [7]. MHS showed a positive association with age among participants with early DS patients but not in participants with late DS, suggesting that older individuals may exhibit greater psychological resilience. However, this association was not observed in late DS, where age did not confer similar psychological benefits, possibly due to the cumulative burden of symptoms and reduced quality of life in later stages. Strong correlations between TIN scores and overall HRQoL and MHS scores were observed for early and late DS, with higher TIN scores significantly predicting lower MHS (p=0.018). Female participants and those with higher HHI reported better physical and mental health outcomes, while patients who underwent RYGB surgery or were non-compliant with follow-up visits due to time constraints exhibited significantly lower PHS scores. These findings underscore the clinical importance of monitoring DS severity and patient compliance post-bariatric surgery, to optimize long-term HRQoL outcomes.

The cross-sectional design of this study, like many similar investigations, limits the ability to draw causal inferences, as it only provides a snapshot in time and identifies associations rather than direct cause-and-effect relationships. The use of convenience sampling may introduce selection bias and restrict the generalizability of the results to the broader population. This sampling approach, together with the exclusion of asymptomatic or non-DS participants, means that our

estimates of HRQoL primarily reflect a more symptomatic subset of post-bariatric patients rather than all individuals at risk of DS. Despite these limitations, this research is a valuable addition to the literature, as it is among the first studies in Saudi Arabia to specifically evaluate the impact of early and late DS on quality of life in post-bariatric surgery patients, addressing a notable gap in regional data.

## Conclusion

This study highlights the significant effects of DS on the HRQoL for patients after bariatric surgery. Our findings indicate that the most common symptoms were a desire to lie down (79.3%) and the prevalence of nausea (37.9%). Patients with severe late DS are more inclined to seek help from various healthcare providers, highlighting potential gaps in effective management strategies. Our study also shows a marked decline in both PHS and MHS for those with severe DS symptoms. We found a positive relationship between mental health and age in early DS patients. Overall, this research highlights the critical need for prompt identification of early and late DS symptoms and effective management strategies to enhance HRQoL outcomes for bariatric patients; thus, paving the way for future studies to focus on effective interventions.

## Supporting information

**S1 Table. This is the S1 Table. Descriptive analysis of the patients perceived interference and troubles associated with DS symptoms (n = 237).** This is the S1 Fig legend Note: reported values are frequency (%) or Mean ± SD. [C] **chi-square test,** [F] **Fisher's Exact test,** [T] **independent samples t-test,** [W] **Welch t-test (for unequal variances)** [1] **None-very low: No symptom that's troublesome** [2] **Moderate: at least one symptom that is little to a bit troublesome** [3] **Severe: at least two symptoms with quite a bit to very much troublesome.**
(DOCX)

## Acknowledgments

Special thanks for support by the College of Medicine Research Center, Deanship of Scientific Research, King Saud University Riyadh, Saudi Arabia. We also acknowledge the use of the generative AI tool ChatGPT to support language review and proofreading in the preparation of some sections of this manuscript.

## Author contributions

**Conceptualization:** Leena R. Baghdadi.

**Data curation:** Hoor K. Aloraini, Razan A. Almohanna, Jana I. Alhazmi, Farah M. Alhalafi, Sarah H. Alotaebe.

**Formal analysis:** Leena R. Baghdadi, Hoor K. Aloraini, Razan A. Almohanna, Jana I. Alhazmi, Farah M. Alhalafi, Sarah H. Alotaebe.

**Funding acquisition:** Leena R. Baghdadi.

**Investigation:** Leena R. Baghdadi, Hoor K. Aloraini, Razan A. Almohanna, Jana I. Alhazmi, Farah M. Alhalafi, Sarah H. Alotaebe.

**Methodology:** Leena R. Baghdadi, Hoor K. Aloraini, Razan A. Almohanna, Jana I. Alhazmi, Farah M. Alhalafi, Sarah H. Alotaebe.

**Project administration:** Leena R. Baghdadi.

**Resources:** Leena R. Baghdadi.

**Software:** Leena R. Baghdadi, Hoor K. Aloraini, Razan A. Almohanna, Jana I. Alhazmi, Farah M. Alhalafi, Sarah H. Alotaebe.

**Supervision:** Leena R. Baghdadi.

**Validation:** Leena R. Baghdadi.

**Visualization:** Leena R. Baghdadi.

**Writing – original draft:** Leena R. Baghdadi, Hoor K. Aloraini, Razan A. Almohanna, Jana I. Alhazmi, Farah M. Alhalafi, Sarah H. Alotaebe.

**Writing – review & editing:** Leena R. Baghdadi, Hoor K. Aloraini, Razan A. Almohanna, Jana I. Alhazmi, Farah M. Alhalafi, Sarah H. Alotaebe.

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
