## [Decision Letter · Decision Letter 0]

26 Dec 2025

Dear Dr. Baghdadi,

Thank you for submitting your manuscript to PLOS ONE. After careful consideration, we feel that it has merit but does not fully meet PLOS ONE’s publication criteria as it currently stands. Therefore, we invite you to submit a revised version of the manuscript that addresses the points raised during the review process.

I thank the authors for their diligent work in preparing this manuscript. The study has several merits; however, some points require clarification before it can be considered for acceptance. I look forward to receiving a revised version in which the reviewers’ comments are adequately addressed and corrected.

I have 2 minor comments:

There is an inconsistency regarding the study period. In the “Study sample” section, data collection is reported as February–June 2024, whereas the Results section states February–December 2024. Please clarify the actual study period and correct this discrepancy throughout the manuscript.

The figures might need to be submitted in higher resolution for publication.

We look forward to receiving your revised manuscript.

Kind regards,

M Saad Saumtally

Academic Editor

PLOS One

**Journal Requirements:**

1. When submitting your revision, we need you to address these additional requirements. Please ensure that your manuscript meets PLOS ONE's style requirements, including those for file naming. The PLOS ONE style templates can be found at https://journals.plos.org/plosone/s/file?id=wjVg/PLOSOne_formatting_sample_main_body.pdf and https://journals.plos.org/plosone/s/file?id=ba62/PLOSOne_formatting_sample_title_authors_affiliations.pdf 2. Thank you for stating the following in the Acknowledgments Section of your manuscript: Special thanks for support by the College of Medicine Research Center, Deanship of Scientific Research, King Saud University Riyadh, Saudi Arabia. We also acknowledge the use of the generative AI tool ChatGPT to support language review and proofreading in the preparation of some sections of this manuscript. We note that you have provided funding information that is not currently declared in your Funding Statement. However, funding information should not appear in the Acknowledgments section or other areas of your manuscript. We will only publish funding information present in the Funding Statement section of the online submission form. Please remove any funding-related text from the manuscript and let us know how you would like to update your Funding Statement. Currently, your Funding Statement reads as follows: The author(s) received no specific funding for this work.  Please include your amended statements within your cover letter; we will change the online submission form on your behalf. 3. Please provide a complete Data Availability Statement in the submission form, ensuring you include all necessary access information or a reason for why you are unable to make your data freely accessible. If your research concerns only data provided within your submission, please write "All data are in the manuscript and/or supporting information files" as your Data Availability Statement. 4. Please upload a new copy of Figure 1 as the detail is not clear. Please follow the link for more information:  https://journals.plos.org/plosone/s/figures 5. Please include captions for your Supporting Information files at the end of your manuscript, and update any in-text citations to match accordingly. Please see our Supporting Information guidelines for more information: http://journals.plos.org/plosone/s/supporting-information. 6. If the reviewer comments include a recommendation to cite specific previously published works, please review and evaluate these publications to determine whether they are relevant and should be cited. There is no requirement to cite these works unless the editor has indicated otherwise. 

Reviewers' comments:

Reviewer's Responses to Questions

**Comments to the Author**

1. Is the manuscript technically sound, and do the data support the conclusions?

Reviewer #1: Yes

Reviewer #2: Partly

2. Has the statistical analysis been performed appropriately and rigorously?

Reviewer #1: Yes

Reviewer #2: I Don't Know

3. Have the authors made all data underlying the findings in their manuscript fully available?

Reviewer #1: Yes

Reviewer #2: Yes

4. Is the manuscript presented in an intelligible fashion and written in standard English?

Reviewer #1: Yes

Reviewer #2: Yes

**Reviewer #1:** The study evaluated the impact of early and late dumping syndrome (DS) on health-related quality of life (HRQoL) in patients undergoing bariatric surgery in Saudi Arabia, including both sleeve gastrectomy (SG) and Roux-en-Y gastric bypass (RYGB). Considering the epidemic proportions of obesity and the need for its treatment, it brings necessary and important information in the postoperative management of patients. The results revealed a significant impairment of physical and mental health, with low mean scores of the physical (PHS) and mental (MHS) components. The severity of symptoms was a major determinant of the decrease in HRQoL, with patients with severe DS presenting significantly lower scores compared to those with moderate forms. In both early DS and late DS, over 60% of participants reported severe symptoms, the most common being the need to lie down and nausea. The differences in clinical manifestation between the two types of DS influenced the behavior of seeking medical care, with patients with late DS requesting non-surgical consultations more frequently, probably due to persistent neuroglycopenic symptoms. The study provides valuable regional data and emphasizes the need for early identification and effective management of DS to improve postoperative quality of life.

**Reviewer #2:**  Dear Authors,

I read your manuscript carefully. I have some questions and comments and I hope addressing them, can increase the quality of your upcoming manuscript.

1- There is a mismatch between methods (February 2024 until June 2024) and results (between February 2024 and December 2024) in data collection times. Please correct.

2- You mentioned that "Participants without DS (n = 115) were excluded entirely from the analysis." It may prone your study to the selection bias and limit comparing HRQoL between DS and non-DS patients, and make DS as an independent factor for HRQoL questionnaire.

3- The Results indicate no significant difference between early and late DS in several analyses, so please avoid overstating subgroup differences.

4- I suggest using a similar study to assess the severity of DS after RYGB in the discussion part(PMID: 36607445).

**Do you want your identity to be public for this peer review?** For information about this choice, including consent withdrawal, please see our Privacy Policy

Reviewer #1: No

Reviewer #2: No

---

## [Author Response · Author response to Decision Letter 1]

9 Jan 2026

Response letter:

Journal Requirements:

Response:

Thank you for your comment. The file names were updated as requested. The manuscript was edited and formated by a proffesional editor and the journal’s templet was used (please see attched editing certificate).

Special thanks for support by the College of Medicine Research Center, Deanship of Scientific Research, King Saud University Riyadh, Saudi Arabia. We also acknowledge the use of the generative AI tool ChatGPT to support language review and proofreading in the preparation of some sections of this manuscript.

Response:

Thank you for your comment. This study did not receive any funding. Therefore, there is no need to change the funding statement in the manuscript.

3. Please provide a complete Data Availability Statement in the submission form, ensuring you include all necessary access information or a reason for why you are unable to make your data freely accessible. If your research concerns only data provided within your submission, please write " All data are in the manuscript and/or supporting information files " as your Data Availability Statement.

Response:

Thenak youu for your comment. The satemnt “All data are in the manuscript and/or supporting information files” was added as requested on page no. 19, lines 396-397.

4. Please upload a new copy of Figure 1 as the detail is not clear. Please follow the link for more information: https://journals.plos.org/plosone/s/figures

Response:

Thank you for your comment. Figure 1 and 2 were enhanced and uploaded.

Response:

Thank you for your comment. The supporting information section was added in the manuscript on page no.25, lines 525-532 and the in-text citations were updated accordingly.

Response:

Thank you for your comment. References were updated after considering the reiewer’s comment.

Response:

Thank you for your comment. The reference list was updated.

Comments to the Author

1. Is the manuscript technically sound, and do the data support the conclusions?

Reviewer #1: Yes

Reviewer #2: Partly

2. Has the statistical analysis been performed appropriately and rigorously?

Reviewer #1: Yes

Reviewer #2: I Don't Know

3. Have the authors made all data underlying the findings in their manuscript fully available?

Reviewer #1: Yes

Reviewer #2: Yes

4. Is the manuscript presented in an intelligible fashion and written in standard English?

Reviewer #1: Yes

Reviewer #2: Yes

5. Review Comments to the Author

Reviewer #1: The study evaluated the impact of early and late dumping syndrome (DS) on health-related quality of life (HRQoL) in patients undergoing bariatric surgery in Saudi Arabia, including both sleeve gastrectomy (SG) and Roux-en-Y gastric bypass (RYGB). Considering the epidemic proportions of obesity and the need for its treatment, it brings necessary and important information in the postoperative management of patients. The results revealed a significant impairment of physical and mental health, with low mean scores of the physical (PHS) and mental (MHS) components. The severity of symptoms was a major determinant of the decrease in HRQoL, with patients with severe DS presenting significantly lower scores compared to those with moderate forms. In both early DS and late DS, over 60% of participants reported severe symptoms, the most common being the need to lie down and nausea. The differences in clinical manifestation between the two types of DS influenced the behavior of seeking medical care, with patients with late DS requesting non-surgical consultations more frequently, probably due to persistent neuroglycopenic symptoms. The study provides valuable regional data and emphasizes the need for early identification and effective management of DS to improve postoperative quality of life.

Response:

Thank you for your encourging comments.

Reviewer #2: Dear Authors,

I read your manuscript carefully. I have some questions and comments and I hope addressing them, can increase the quality of your upcoming manuscript.

1- There is a mismatch between methods (February 2024 until June 2024) and results (between February 2024 and December 2024) in data collection times. Please correct.

Response:

We thank the reviewer for their invaluable input. The method section was reviweed and the duration was corrected to be from February 2024 to December 2024 (page no.4, line 87).

2- You mentioned that "Participants without DS (n = 115) were excluded entirely from the analysis." It may prone your study to the selection bias and limit comparing HRQoL between DS and non-DS patients, and make DS as an independent factor for HRQoL questionnaire.

Response:

We appreciate the reviewer’s thoughtful comment regarding potential selection bias. Our primary objective was to investigate the determinants and gradients of HRQoL within the DS population (e.g., how the severity and timing of early vs. late DS to specific QoL domains), rather than to compare DS and non-DS patients or to establish DS as an independent determinant of HRQoL at the population level.

We agree that excluding participants without DS (n = 115) restricts formal comparison of HRQoL between DS and non-DS groups and may limit external generalizability. This is now acknowledged as a limitation in the manuscript (page no. 18, lines 372-374). Nonetheless, the observed HRQoL impairments in our symptomatic DS cohort are aligned with prior studies showing that patients with moderate-to-severe dumping symptoms have substantially lower HRQoL scores than asymptomatic or minimally symptomatic post-gastrectomy controls, supporting the clinical relevance of our within-DS findings.

3- The Results indicate no significant difference between early and late DS in several analyses, so please avoid overstating subgroup differences.

Response:

Thank you for your comment. The results were reviewed and the overstating sentence comparing early and late DS was deleted (page no.11 , lines 225-226).

4- I suggest using a similar study to assess the severity of DS after RYGB in the discussion part(PMID: 36607445).

Response:

We thank the reviewer for highlighting this study (PMID: 36607445). The main objective of that paper was to evaluate the effect of dumping syndrome (DS) severity on weight loss outcomes after Roux-en-Y gastric bypass (RYGB) in patients with class III obesity. In contrast, our study focuses on assessing health-related quality of life among adult patients in Saudi Arabia with varying DS severity after either sleeve gastrectomy or RYGB.

Given these fundamental differences in primary outcomes (weight loss vs quality of life), patient population, and inclusion of two different bariatric procedures in our work, we believe the study is not directly comparable as a methodological model for our research. However, we agree it remains relevant to cite in the discussion as supportive evidence that DS severity can have clinically meaningful consequences after bariatric surgery, while emphasizing that our study extends this literature by addressing patient-reported quality of life across different surgical techniques in a Middle Eastern population (page no. 17, lines 350-352).

6. PLOS authors have the option to publish the peer review history of their article (what does this mean?). If published, this will include your full peer review and any attached files.

Do you want your identity to be public for this peer review? For information about this choice, including consent withdrawal, please see our Privacy Policy.

Reviewer #1: No

Reviewer #2: No

---

## [Decision Letter · Decision Letter 1]

10 Mar 2026

The effect of dumping syndrome severity on health-related quality of life among bariatric surgery patients in Saudi Arabia

PONE-D-25-62832R1

Dear Dr. Baghdadi,

We’re pleased to inform you that your manuscript has been judged scientifically suitable for publication and will be formally accepted for publication once it meets all outstanding technical requirements.

Kind regards,

M Saad Saumtally

Academic Editor

PLOS One

Reviewers' comments:

Reviewer's Responses to Questions

**Comments to the Author**

Reviewer #1: (No Response)

Reviewer #2: All comments have been addressed

2. Is the manuscript technically sound, and do the data support the conclusions?

Reviewer #1: Yes

Reviewer #2: Yes

3. Has the statistical analysis been performed appropriately and rigorously?

Reviewer #1: Yes

Reviewer #2: I Don't Know

4. Have the authors made all data underlying the findings in their manuscript fully available?

Reviewer #1: Yes

Reviewer #2: Yes

5. Is the manuscript presented in an intelligible fashion and written in standard English?

Reviewer #1: Yes

Reviewer #2: Yes

Reviewer #1: This study represents a valuable and timely contribution to the literature addressing the impact of dumping syndrome (DS) on health-related quality of life (HRQoL) in patients following bariatric surgery. One of its principal strengths lies in the differentiated assessment of early and late forms of DS, as well as in the evaluation of symptom severity in relation to physical (PHS) and mental (MHS) health scores. This stratified approach allows for a more nuanced understanding of how symptom intensity translates into patient-reported outcomes.

The inclusion of a Saudi population addresses an important regional gap in the literature, enhancing the epidemiological and contextual relevance of the findings. Furthermore, the consistency of the results with those of the authors’ previous study strengthens the scientific coherence of the work and suggests reproducibility of the observed patterns within this population. The use of multivariable analysis to identify DS as an independent negative predictor of HRQoL represents a solid methodological element that reinforces the robustness of the conclusions.

Another noteworthy aspect is the evaluation of healthcare-seeking behavior and the role of preoperative patient education, both of which carry direct clinical implications. The observed association between preoperative education and improved mental health scores, although of borderline statistical significance, highlights the potential value of structured educational interventions in optimizing postoperative outcomes.

Although the cross-sectional design and convenience sampling limit causal inference and generalizability, these constraints are appropriately acknowledged by the authors. Overall, the study provides clinically meaningful insights, underscores the importance of systematic monitoring of DS severity, and supports the need for prospective research to further strengthen the evidence base and guide effective therapeutic strategies.

Reviewer #2: Dear Authors,

Thank you for addressing all my comments. I satisfied by your responses and accept your study limitations. I think it can be publishable now.

**Do you want your identity to be public for this peer review?** For information about this choice, including consent withdrawal, please see our Privacy Policy

Reviewer #1: No

Reviewer #2: **Yes:** Mohammad Kermansaravi

---

## [Editor Report · Acceptance letter]

PONE-D-25-62832R1

PLOS One

Dear Dr. Baghdadi,

I'm pleased to inform you that your manuscript has been deemed suitable for publication in PLOS One. Congratulations! Your manuscript is now being handed over to our production team.

Kind regards,

on behalf of

Dr. M Saad Saumtally

Academic Editor

PLOS One